# First 24-Hour Potassium Concentration and Variability and Association with Mortality in Patients Requiring Continuous Renal Replacement Therapy in Intensive Care Units: A Hospital-Based Retrospective Cohort Study

**DOI:** 10.3390/jcm11123383

**Published:** 2022-06-13

**Authors:** I-Chieh Mao, Pei-Ru Lin, Shin-Hwar Wu, Hsin-Hui Hsu, Pei-Shan Hung, Chew-Teng Kor

**Affiliations:** 1Division of Critical Care Internal Medicine, Department of Emergency Medicine and Critical Care, Changhua Christian Hospital, Changhua 500, Taiwan; maoichieh@gmail.com (I.-C.M.); 126366@cch.org.tw (S.-H.W.); 117610@cch.org.tw (H.-H.H.); 181006@cch.org.tw (P.-S.H.); 2Big Data Center, Changhua Christian Hospital, Changhua 500, Taiwan; 183778@cch.org.tw; 3Graduate Institute of Statistics and Information Science, National Changhua University of Education, Changhua 500, Taiwan

**Keywords:** continuous renal replacement therapy, potassium target, potassium variability, mortality, ICU, critically ill patient

## Abstract

Serum potassium (K+) levels between 3.5 and 5.0 mmol/L are considered safe for patients. The optimal serum K+ level for critically ill patients with acute kidney injury undergoing continuous renal replacement therapy (CRRT) remains unclear. This retrospective study investigated the association between ICU mortality and K+ levels and their variability. Patients aged >20 years with a minimum of two serum K+ levels recorded during CRRT who were admitted to the ICU in a tertiary hospital in central Taiwan between January 01, 2010, and April 30, 2021 were eligible for inclusion. Patients were categorized into different groups based on their mean K+ levels: <3.0, 3.0 to <3.5, 3.5 to <4.0, 4.0 to <4.5, 4.5 to <5.0, and ≥5.0 mmol/L; K+ variability was divided by the quartiles of the average real variation. We analyzed the association between the particular groups and in-hospital mortality by using Cox proportional hazard models. We studied 1991 CRRT patients with 9891 serum K+ values recorded within 24 h after the initiation of CRRT. A J-shaped association was observed between serum K+ levels and mortality, and the lowest mortality was observed in the patients with mean K+ levels between 3.0 and 4.0 mmol/L. The risk of in-hospital death was significantly increased in those with the highest variability (HR and 95% CI = 1.61 [1.13–2.29] for 72 h mortality; 1.39 [1.06–1.82] for 28-day mortality; 1.43 [1.11–1.83] for 90-day mortality, and 1.31 [1.03–1.65] for in-hospital mortality, respectively). Patients receiving CRRT may benefit from a lower serum K+ level and its tighter control. During CRRT, progressively increased mortality was noted in the patients with increasing K+ variability. Thus, the careful and timely correction of dyskalemia among these patients is crucial.

## 1. Introduction

With an increase in the incidence of acute kidney injury (AKI) in recent decades, AKI is now emerging as a global health-care concern. Up to 60% of patients admitted to the intensive care unit (ICU) develop AKI, which may be associated with complications such as fluid overload, refractory hyperkalemia, and metabolic derangement [1]. In addition, AKI is associated with adverse outcomes including a prolonged stay in the ICU and hospital, higher mortality rates, and subsequent development of chronic kidney disease (CKD). The Acute Kidney Injury-Epidemiologic Prospective Investigation study [2] reported that up to 23.5% of patients with AKI admitted to the ICU and 13.5% of all ICU patients required renal replacement therapy (RRT).

Continuous RRT (CRRT) is a type of RRT that provides the advantages of gentle waste and fluid removal and stable acid–base and electrolyte correction. Therefore, CRRT is the most commonly used for hemodynamically unstable patients with AKI in the ICU [2]. Despite major advances in the management of critically ill patients with AKI, the mortality rate remains high, ranging from 28% to 90% [3,4,5,6,7]. Several factors can predict mortality in patients with AKI after the initiation of CRRT in the ICU, including demographic characteristics and variables that denote the severity of illness.

Potassium (K+) is a crucial electrolyte that plays a vital role in maintaining normal cell function through the kidneys [8,9]. Therefore, K+ homeostasis is critical, and its imbalance can lead to differential disease progression and subsequent life-threatening complications [9], especially in critically ill patients [10]. To date, the association of K+ levels and their variability with clinical outcomes has mainly been investigated in patients with advanced CKD and peritoneal dialysis [11,12]. A large-scale study of the German Clinical Trials Register reported that increased serum K+ levels and their variability during hospitalization were associated with increased in-hospital mortality in ICU patients [13]. The conditions of ICU patients are diverse, and target K+ ranges may vary depending on the patient’s condition. To the best of our knowledge, no study has examined the association of serum K+ levels and their variability with mortality in patients undergoing CRRT.

We hypothesized that K+ levels can be used not only for clinical management but also as prognostic factors for patient outcomes. Thus, we conducted this study to evaluate the association of K+ levels and their variability within 24 h after the initiation of CRRT with mortality in patients undergoing CRRT.

## 2. Materials and Methods

### 2.1. Study Population

This retrospective observational cohort study was conducted at Changhua Christian Hospital (CCH), a tertiary medical center with a total of 130 ICU beds in five separate wards in central Taiwan. A total of 3303 consecutive ICU patients who received CRRT between 1 January 2010, and 30 April 2021, were identified based on the data retrieved from the CCH Clinical Research Database (CCHRD), which contains information from all electronic medical record systems, including data on CRRT, inpatient care, prescriptions, laboratory results, clinical visits, and death records.

To examine the effect of K+ variability resulting from CRRT, only patients receiving CRRT for >24 h were studied. Patients aged >20 years with a minimum of two serum K+ values recorded during CRRT who underwent treatment in a medical or surgical ICU were eligible for inclusion. Patients with pre-existing ESRD and incomplete biochemical data were excluded. Finally, 1991 patients were enrolled in this study (Figure 1).

### 2.2. CRRT Protocol

The initiation of CRRT is at the discretion of the intensivist, and continuous venovenous hemofiltration is the modality most commonly chosen in our institution. Due to the complexity of CRRT in patients with AKI, the initiation of CRRT is guided by KDIGO Clinical Practice Guideline [14] and depends on clinical judgment. In most cases, femoral venous catheters were chosen as vascular access for CRRT and Infomed HF-440 with polyethersulfone (infomed polyethersulfone DF40, Geneva, Switzerland) hollow-fiber hemocircuits were applied for treatment. Most of the dialysate used during CRRT was the Prismasol B0 (Baxter) dialysis solution, which contains no potassium in the reconstituted solution. The potassium was used according to a predefined protocol (Table A1). The initial flow rate was 100 to 150 mL/h, and it gradually increased according to hemodynamic status. The target clearance was greater than 20 mL/kg/h in most patients and increased if possible. Additionally, anticoagulation therapy was administered if there was no bleeding tendency. To ensure the safety and efficacy of CRRT, physicians and experienced nurses discussed the body weight, urine output, laboratory results, actual dose delivered and hemodynamic status after CRRT was started.

### 2.3. K+ Measurements and Other Confounders

The exposure of interest was the patients’ 24 h mean K+ value after receiving CRRT. The patients were categorized into different groups on the basis of their mean K+ levels: <3.0, 3.0 to <3.5, 3.5 to <4.0, 4.0 to <4.5, 4.5 to <5.0, and ≥5.0 mmol/L. K+ variability was measured using the average real variation (ARV) of each patient’s serum K+ measurements, which is a novel measure proposed by Mena et al. [15] that represents short-term, intertest variability in K+ in patients receiving CRRT. The ARV considers the order of measurements and quantifies the differences between two readings; the ARV corrects for the limitation of the standard deviation and accounts only for the dispersion of values around the mean [16,17]. The ARV was calculated using the following formula:(1)$$\mathrm{ARV}=\frac{1}{\sum w_{i}}\sum_{i=1}^{n-1}w_{i}\times \left|K_{i+1}-K_{i}\right|$$
where *n* is the number of serum K+ records, *i* ranges from 1 to *n* − 1, and *w**_i_* is the time interval between K*_i_* and K*_i_*_+1_.

K+ variability was divided by quartiles, and high variability was defined as the presence of values in the highest quartile. Finally, we computed the risk matrix of the mean K+ level and its variability and analyzed in-hospital mortality risk.

The following confounders were examined: age, sex, BMI, Acute Physiology and Chronic Health Evaluation (APACHE) II scores at ICU admission, AKI diagnostic criteria, the timing of CRRT initiation, vital signs within 24 h after CRRT initiation, urine output, medication use and multiple organ support before CRRT, serum biochemical data (albumin, hemoglobin, WBC count, platelet count, pH, sodium, lactate, calcium, base excess, O_2_ saturation, and creatinine), and co-morbidities. The diagnosis of AKI was based on Kidney Disease Improving Global Outcomes (KDIGO) criteria for serum creatinine elevation, which was determined by comparing baseline serum creatinine levels on admission with serum creatinine levels before the initiation of CRRT. All confounders were obtained from CCHRD.

### 2.4. Endpoint

The death records of the patients were reviewed, and the primary endpoint was in-hospital mortality after the patients received CRRT. The secondary endpoint was 72 h, 28-day, and 90-day mortality after the patients received CRRT. The follow-up of mortality risk was initiated at CRRT, and the data of patients were censored at the end of the respective follow-up periods (at discharge or 72 h or 28-day and 90-day follow-ups) or the last date of available follow-up data, whichever occurred first.

### 2.5. Statistical Analysis

Categorical and continuous variables are expressed as numbers (proportions) and medians and interquartile ranges, respectively. The chi-square test and Kruskal–Wallis H test were performed to compare categorical and continuous variables, respectively. Mortality rates (per 100 patient-days) during the follow-up period according to the K+ level and its variability within 24 h after receiving CRRT are presented in Figure 2. Restricted cubic splines were used to visualize the hazard ratio (HR) of mortality for the mean K+ level and its variability. Survival analysis was performed to evaluate the association of the K+ level and its variability with mortality; the categories of 3.0 to <3.5 mmol/L and 0.35 to <0.50 mmol/L were used as reference groups for the K+ level and its variability, respectively. Crude and multivariate Cox’s proportional hazard models were used to estimate mortality rates during the follow-up period according to the K+ level and its variability categories. This study investigated the association of the K+ level and its variability with survival outcomes; the results would help clinicians manage K+ levels to prevent death. Thus, the risk matrix provides an easy-to-understand method to visualize the results of the additive effects of mean K+ levels and their variability as well as their association with mortality. In fact, high lactate levels were found in the group with a mean serum K+ < 3.0 in our study. To improve the robustness of our results, a sensitivity analysis was performed for patients with pH < 7.3 and lactate > 5, and a total of 299 patients were analyzed.

All statistical analyses were performed using SAS, and a visualization plot was plotted using R software (version 4.1.0; The Comprehensive R Archive Network: http://cran.r-project.org, accessed on 18 May 2021). All two-sided *p* values of <0.05 were considered statistically significant.

## 3. Results

### 3.1. Baseline Characteristics of the Study Cohort

We studied 1991 patients who received CRRT, of whom 1346 (67.6%) died in the hospital. A total of 9891 serum K+ values were recorded within 24 h after the initiation of CRRT in these patients. The median K+ level and its median variability per patient were 3.6 and 0.5 mmol/L, respectively, and the median number of (K+) measurements was 5. The median age of the patients was 70 years. Of the 1991 patients, 1267 (63.6%) were men, and 1561 (78.4%) were treated in the ICU. Furthermore, 575 (28.9%) patients had oliguria, 452 (22.7%) had anuria, and 501 (25.2%) had AKI with KDIGO-defined serum creatinine elevation, and 463 (23.3%) had other causes.

Table 1 lists the baseline characteristics of the patients stratified by K+ levels. The patients with higher mean K+ levels had higher K+ variability, WBC counts, platelet counts, pH values, and creatinine levels as well as a higher prevalence of CKD, cardiac arrhythmia occurrence and insulin use. The patients with a mean K+ level of 4.0 to < 4.5 mmol/L were older and had the highest prevalence of diabetes mellitus. Moreover, the patients with a mean K+ level of 3.0 to < 4.0 mmol/L were more likely to use extracorporeal membrane oxygenation. The patients with a mean K+ level of 4.5 to < 5.0 mmol/L had the lowest urine output. The lowest mortality rates were observed in the patients with mean K+ levels between 3.0 and < 3.5 mmol/L.

### 3.2. Association of Mean K+ Levels after CRRT Initiation with Mortality

As shown in Figure 2a, the lowest and highest incidence of death (72 h, 28-day, 90-day and in-hospital death) was respectively observed in the patients with K+ levels of 3.0 to <3.5 mmol/L (7.7, 29.4, 15.9 and 23.9 deaths per 100 patient-days, respectively) and the patients with K+ levels of ≥5.0 mmol/L (23.8, 63.6, 27.4 and 42.4 deaths per 100 patient-days, respectively). The mortality rates were slightly higher in the patients with K+ levels of <3.0 mmol/L (12.3, 38.8, 20.9, and 30.4 deaths per 100 patient-days, respectively) than in those with K+ levels between 3.0 and <3.5 mmol/L.

In the unadjusted model, the serum K+ levels of 4.0 to <4.5, 4.5 to <5.0, and ≥5.0 mmol/L were associated with an increased risk of mortality; the serum K+ levels of <3.0 mmol/L were associated with marginally increased mortality risk (Table A2). After adjustment for confounders, the risk of in-hospital death was significant in those with K+ levels of 4.0 to <4.5, 4.5 to <5.0, and ≥5.0 mmol/L (HR and 95% confidence interval [CI] = 1.59 [1.31–1.94], 1.85 [1.44–2.36], and 1.72 [1.30–2.27], respectively). The risk of 72 h, 28-day, and 90-day mortality was significantly higher in those with K+ levels of 4.0 to <4.5, 4.5 to <5.0, and ≥ 5.0 mmol/L (HR and 95% CI = 1.63 [1.17–2.26], 2.12 [1.46–3.09], and 2.52 [1.69–3.77] for 72 h mortality, respectively; HR and 95% CI = 1.65 [1.32–2.07], 2.00 [1.52–2.64], and 2.10 [1.54–2.86] for 28-day mortality, respectively, and HR and 95% CI = 1.66 [1.35–2.04], 2.06 [1.60–2.66], and 1.96 [1.45–2.64] for 90-day mortality, respectively). By contrast, the patients with K+ levels of <3.0 mmol/L exhibited an increased tendency of mortality, although the finding was not statistically significant (Table 2). As shown in Figure 3a,c,e,g, restricted cubic splines for the association of mean serum K+ levels with mortality as continuous variables were J-shaped. The mortality risk was the lowest for the serum K+ levels of approximately 3.0 to <3.5 mmol/L and steadily increased with serum K+ levels and slightly increased at K+ levels or <3.0 mmol/L; this finding is consistent with the results of the multivariate Cox model.

### 3.3. Association of K+ Variability after CRRT Initiation with Mortality

As shown in Figure 2b, an increased risk of mortality was not observed in the patients with K+ variability of <0.78 mmol/L (Q1 to Q3), whereas the patients with K+ variability of >0.78 mmol/L (Q4) had the highest mortality rates (21.7, 59.3, 30.9, and 41.4 deaths per 100 patient-days, respectively). In the unadjusted model, the highest K+ variability (Q4) was associated with an increased risk of mortality (Table A2). After adjustment for confounders, the risk of in-hospital death was significantly higher in those with the highest K+ variability (HR and 95% CI = 1.61 [1.13–2.29] for 72 h mortality; 1.39 [1.06–1.82] for 28-day mortality; 1.43 [1.11–1.83] for 90-day mortality, and 1.31 [1.03–1.65] for in-hospital mortality, respectively). As shown in Figure 3b,d,f,h, the risk of mortality was lower in the patients with a K+ variability of <0.78 mmol/L, whereas the risk of mortality steadily increased in those with a K+ variability of >0.78 mmol/L (Table 2).

### 3.4. In-Hospital Mortality Risk Matrix of Mean (K+) Levels and Their Variability

Figure 4 presents the mortality risk matrix determined using the mean K+ level categories combined with K+ variability quantiles. The patients with higher mean K+ levels and concomitant larger K+ variability had a higher risk of mortality (Figure 4). In the patients with serum K+ levels of <4.0 mmol/L, changes in K+ levels did not increase the risk of mortality, except in those with serum K+ levels of <3.0 mmol/L combined with K+ variability of <0.35 mmol/L and those with serum K+ levels of 3.5 to <4.0 mmol/L combined with K+ variability of >0.78. The patients with serum K+ levels of ≥4.0 mmol/L had a higher risk of mortality regardless of the change in K+ values.

Figure A1 shows the mortality risk matrix used for sensitivity analysis, and patients with serum K+ levels < 3.0 mmol/L and changes in K+ levels < 0.35 mmol/L were also associated with a higher risk of death. Furthermore, the results were consistent with those of the preliminary analysis.

### 3.5. Other Significant Factors Affecting In-Hospital Mortality

Figure 5 presents the association between other factors and in-hospital mortality. A significantly higher risk of in-hospital mortality was associated with vasopressor use (HR and 95% CI = 1.25 [1.01–1.56]), higher lactate levels (HR and 95% CI = 1.05 [1.03–1.06]), higher respiratory rates (HR and 95% CI = 1.03 [1.01–1.05]), higher pulse rates (HR and 95% CI = 1.01 [1.00–1.01]), higher APACHE II scores (HR and 95% CI = 1.02 [1.02–1.03]), older age (HR and 95% CI = 1.01 [1.01–1.02]), and delayed CRRT initiation (HR and 95% CI = 1.57 [1.36–1.82]). A lower risk of in-hospital mortality was associated with higher creatinine levels (HR and 95% CI = 0.96 [0.93–0.98]), higher O2 saturation (HR and 95% CI = 0.98 [0.97–0.99]), higher base excess (HR and 95% CI = 0.98 [0.97–1.00]), higher platelet counts (HR and 95% CI = 0.99 [0.98–1.00]), higher albumin levels (HR and 95% CI = 0.82 [0.74–0.90]), pre-existing diabetes mellitus (HR and 95% CI = 0.82 [0.71–0.94]), and higher systolic blood pressure (HR and 95% CI = 0.99 [0.99–1.00]). A body temperature of over 36.2 °C was marginally protective against death (HR and 95% CI = 0.87 [0.75–1.01]).

## 4. Discussion

The results of this study revealed a strong J-shaped association between serum K+ levels and ICU mortality, and patients with mean K+ levels between 3.0 and 4.0 mmol/L had the lowest mortality rate. In-hospital and ICU mortality were significantly higher in the patients with hyperkalemia (mean serum K+ levels of ≥4.0 mmol/L) but increased nonsignificantly in the patients with hypokalemia (mean serum K+ levels of <3.0 mmol/L). Furthermore, a higher K+ variability within the first 24 h after the initiation of CRRT resulted in increased mortality. In addition, we observed that CRRT initiation 24 h after renal failure increased the mortality risk.

Whether mean K+ levels and K+ variability are associated with mortality in patients with AKI requiring CRRT in the ICU remains unclear. Currently, a K+ level between 3.5 and 5.0 mmol/L is considered safe for critically ill patients [18]. In a retrospective study conducted by Hessels et al. in 2015 [18], a U-shaped association was observed between K+ levels and in-hospital mortality for all the ICU patients, and the serum K+ levels of 3.5 to 5.0 mmol/L were associated with the lowest mortality rates. However, the optimal range of serum K+ levels for specific patient groups in the ICU remains inconclusive. Studies have reported that K+ levels range from 3.5 to 4.5 mmol/L [19] and from 4.5 to 5.5 mmol/L [20] in patients with acute myocardial infarction and from >3.5 to 4.0 mmol/L in patients with atrial fibrillation [13]. However, in patients with acute respiratory distress syndrome (ARDS) with positive fluid balance, relative hyperkalemia (up to 5.9 mmol/L) was associated with a decreased risk of death [21]. Compared with previous studies, our study demonstrated that the patients with AKI receiving CRRT in the ICU had K+ levels ranging from 3.0 to 4.0 mmol/L. The possible explanation for this finding is that the patients requiring CRRT were often hemodynamically unstable and required intravenous inotropic agents. Inotropic agents, such as norepinephrine, epinephrine, vasopressin, and dopamine, are known to be arrhythmogenic [22]. Both the lower and higher values of serum inotropic agents can exert electrophysiological effects, thus promoting cardiac arrhythmia or myocardial ischemia [23,24]. Thus, patients receiving CRRT in the ICU might be more vulnerable to cardiovascular events and may require a stricter control of serum K+ levels. Thus, K+ levels ranging from 3.5 to 5.0 mmol/L should not be regarded as normal in this critical population. The K+ monitoring and correction protocols for CRRT in each institute may need to be revised to achieve more favorable clinical outcomes.

Recently, the variability or fluctuation of serum K+ levels has emerged as a new focus in the investigation of its relationship with mortality in the hospital setting. In their monocentric and retrospective observational study conducted in Rome, Lombardi et al. examined 64,057 adult hospitalized patients and reported that high K+ variability is an independent risk factor for in-hospital mortality, even in patients with a normal K+ range [25]. However, in this large-scale cohort study, the data were insufficient to determine the relationship in specific medical conditions. Some previous studies have investigated the association in critically ill patients. Using a computerized protocol designed for a surgical ICU [26] to minimize the time for patients with hypokalemia and hyperkalemia, Hessels et al. reported that a low mortality rate was associated with low K+ variability [18]. In the Soroka Acute Myocardial Infarction II (SAMI-II) Project, K+ variability was associated with an increased risk of mortality in patients with acute myocardial infarction [27]. Thongprayoon et al. indicated that hypokalemia (≤3.4 mmol/L) and hyperkalemia (≥4.5 mmol/L) before CRRT and hyperkalemia (≥4.5 mmol/L) during CRRT predicted higher 90-day mortality [28]. However, studies evaluating the prognostic value of K+ fluctuations in patients receiving CRRT are scant. To the best of our knowledge, this study is the first to focus on this topic. Although a direct causal relationship could not be demonstrated in this study, several possible explanations can be provided for the relationship between K+ variability and mortality in our study. First, fluctuations in the resting electrical condition of the cell membrane could increase cellular instability and consequently increase the risk of arrhythmogenic death. Another explanation is that a higher K+ variability may be a surrogate marker of baseline characteristics or disease processes that result in poorer prognosis. Low K+ variability was associated with an increased mortality rate in both hypokalemia and hyperkalemia (Figure 4). Our results suggest that clinicians should correct dyskinesia more intensively to achieve normokalemic status.

In line with the findings of previous studies, we observed lower mortality rates in the patients with higher serum albumin levels [29] and pre-existing diabetes mellitus [30]. By contrast, the delayed initiation of CRRT was associated with an increased risk of mortality, which is similar to the result of the ELAIN randomized clinical trial [31]. In addition, vasopressor therapy and the presence of malignancy were determined as risk factors for mortality in previous studies [32,33]. A marginally lower mortality was observed in the patients with higher body temperature in our study. Hypothermia was reported to have negative clinical consequences [34,35]. In ICU patients, hypothermia risk is increased by sedation, immobility, paralytic drug use, sepsis, underlying endocrine disorders, and higher CRRT dose. However, the patients’ body temperature could partially be manipulated through extracorporeal blood circulation and warming system use in CRRT. Whether body temperature is a maker or indicator of ICU mortality and the optimal range of body temperature during CRRT could not be determined in our study and warrants further investigation.

This study has some limitations. First, because this was a retrospective observational study, the causal relationship between serum K+ levels or their variability and mortality could not be determined. Second, we pooled the data of all adult patients admitted to medical and surgical ICUs together. However, different K+ target ranges may exist depending on individual conditions, and further research is needed to account for the possibility of heterogeneous effects based on baseline characteristics, especially in patients with higher edema, sepsis, and disease severity. Third, some possible residual confounders may not have been considered despite the adjusted analysis conducted in this study. Furthermore, we did not record the exact cause of death to address the mechanistic linkage in our findings. Finally, we conducted this study by using a large electronic database from a single center with a primarily Taiwanese patient population. This design might limit the generalizability of our results to other patient populations.

## 5. Conclusions

Our findings indicate that adult patients with AKI receiving CRRT in the ICU may benefit from a more stringent K+ control target between 3.0 and 4.0 mmol/L within 24 h after the initiation of CRRT. A negative correlation was observed between mortality and K+ variability; however, the causal relationship remains unclear. For the management of dyskalemia, prompt but careful correction with minimal K+ fluctuation is suggested. Hyperkalemia is much more encountered than hypokalemia in AKI patients and is always corrected more aggressively. However, our results remind clinicians of the importance of the management of hypokalemia. For hypokalemia ((K+) < 3.0 mmol/mL) during CRRT, a shorter interval of (K+) measurement may be needed in order to correct the imbalance earlier. Additional large randomized controlled trials are needed to confirm our findings, and their results may change the CRRT protocol in the future.

## Figures and Tables

**Figure 1 jcm-11-03383-f001:**
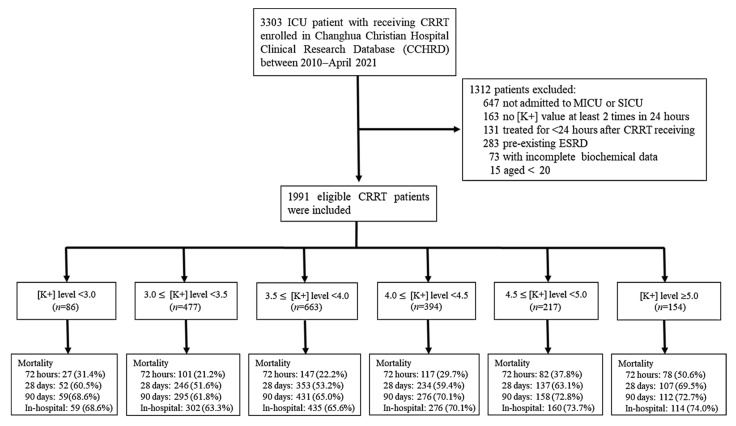
Flowchart of the study population.

**Figure 2 jcm-11-03383-f002:**
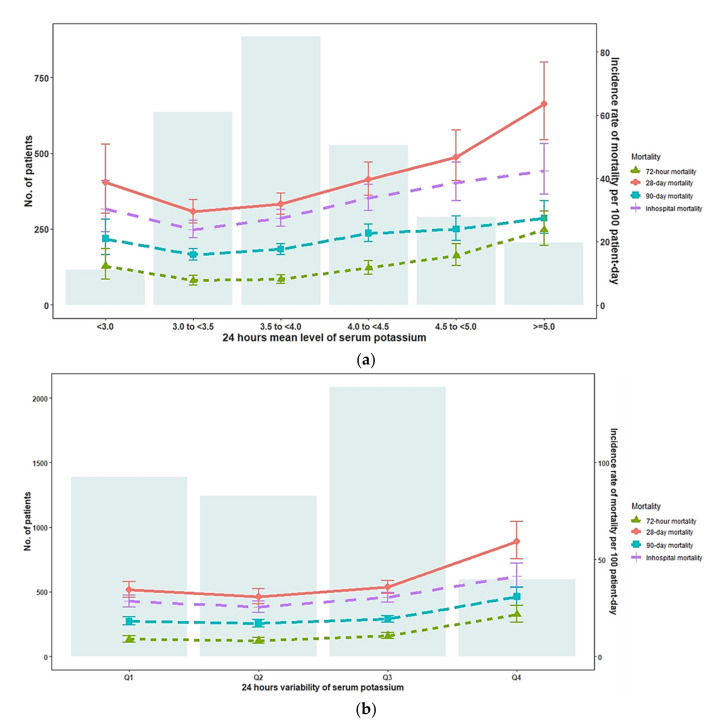
The distribution of serum (K+) level and serum (K+) variability within 24 h after initial CRRT and corresponding mortality rate. (**a**) serum (K+) level within 24 h after initial CRRT; (**b**) serum (K+) variability within 24 h after initial CRRT.

**Figure 3 jcm-11-03383-f003:**
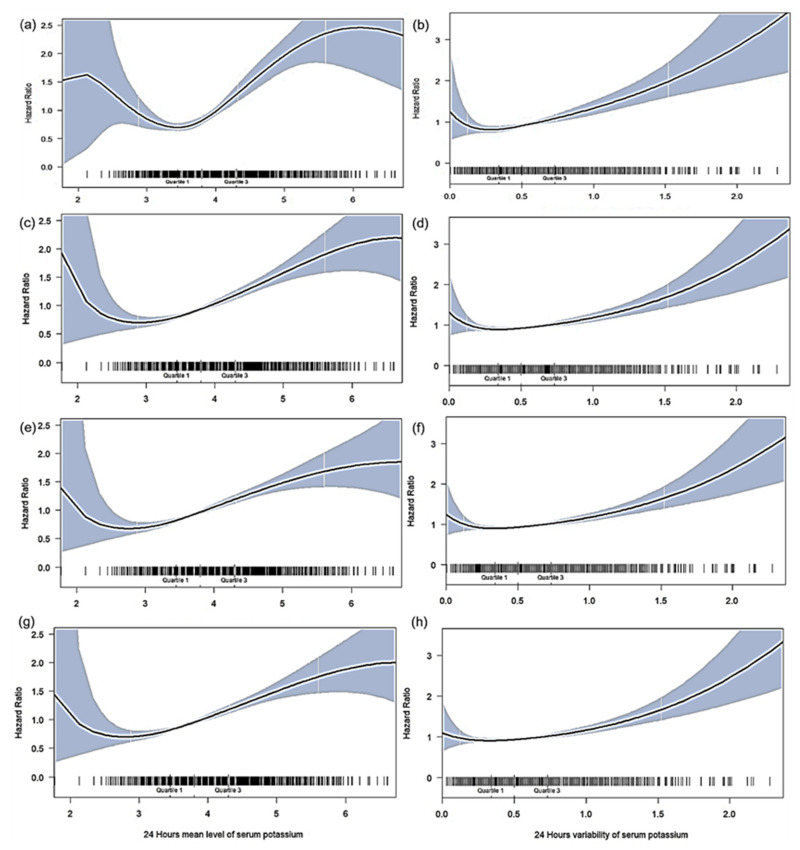
The restricted cubic spline plots depicting the association between serum (K+) level, variability and mortality risk. (**a**) 72-h mortality on serum (K+) level; (**b**) 72-h mortality on serum (K+) variability; (**c**) 28-day mortality on serum (K+) level; (**d**) 28-day mortality on serum (K+) variability; (**e**) 90-day mortality on serum (K+) level; (**f**) 90-day mortality on serum (K+) variability; (**g**) In-hospital mortality on serum (K+) level; (**h**) In-hospital mortality on serum (K+) variability.

**Figure 4 jcm-11-03383-f004:**
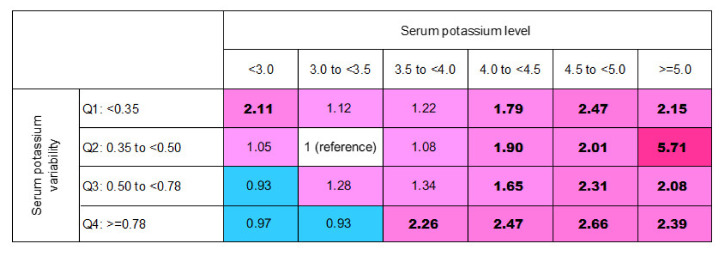
Risk matrices showing the adjusted HRs for in-hospital mortality by using serum potassium level categories and variability in quartiles. The color of the reference cells is white. For HR < 1.0 were represented as blue, while for HR >1.0 was pink. We color the cells from light to dark (away from 1.0). The numbers in bold indicate they are significant (*p* < 0.05).

**Figure 5 jcm-11-03383-f005:**
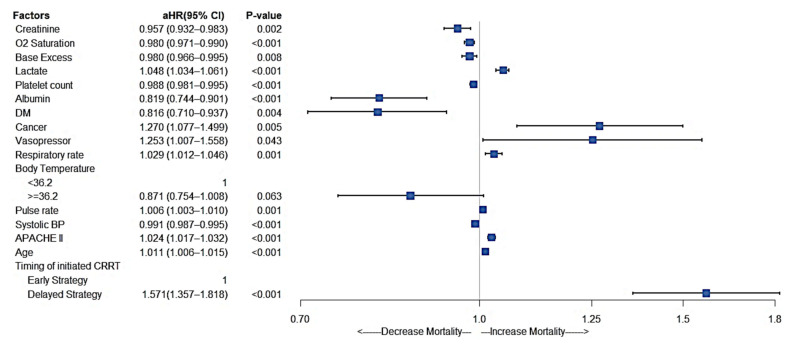
Other significant factors associated with mortality.

**Table 1 jcm-11-03383-t001:** Baseline characteristics of CRRT patients according to categories of serum potassium.

	The Levels of Serum (K+) within 24 h after Initial CRRT						*p* Value
	<3.0	3.0 to <3.5	3.5 to <4.0	4.0 to <4.5	4.5 to <5.0	≥5.0	
Sample size	86	477	663	394	217	154	
Gender, Male	50 (58.1%)	286 (60%)	427 (64.4%)	251 (63.7%)	147 (67.7%)	106 (68.8%)	0.189
Age	66 (56–78)	69 (56–80)	70 (59–80)	73 (62–81)	71 (60–80)	70 (58–82)	0.007
BMI	23.1 (20.3–26.6)	24.4 (21.5–28)	24.6 (21.4–27.5)	24.2 (20.9–28.2)	25 (21.8–27.8)	24 (22.1–28.9)	0.261
APACHE II at admission	29 (23–37)	28 (22–35)	29 (22–35)	29 (22–35)	30 (24–36)	30 (23–38)	0.071
ICU type							
Medicine ICU	54 (62.8%)	355 (74.4%)	531 (80.1%)	319 (81%)	175 (80.6%)	127 (82.5%)	0.001
Surgery ICU	32 (37.2%)	122 (25.6%)	132 (19.9%)	75 (19%)	42 (19.4%)	27 (17.5%)	
The average level of serum (K+) within 24 h after initial CRRT	2.9 (2.7–2.9)	3.3 (3.2–3.4)	3.7 (3.6–3.9)	4.2 (4.1–4.4)	4.7 (4.6–4.9)	5.4 (5.2–5.7)	<0.001
The variability of serum (K+) within 24 h after initial CRRT	0.51 (0.36–0.71)	0.40 (0.26–0.58)	0.46 (0.33–0.66)	0.55 (0.39–0.78)	0.64 (0.46–0.91)	0.78 (0.58–1.11)	<0.001
Q1: <0.35	20 (23.3%)	190 (39.8%)	193 (29.1%)	74 (18.8%)	30 (13.8%)	13 (8.4%)	<0.001
Q2: 0.35 to <0.50	22 (25.6%)	116 (24.3%)	178 (26.8%)	100 (25.4%)	35 (16.1%)	15 (9.7%)	
Q3: 0.50 to <0.78	35 (40.7%)	144 (30.2%)	249 (37.6%)	175 (44.4%)	107 (49.3%)	72 (46.8%)	
Q4: ≥0.78	9 (10.5%)	27 (5.7%)	43 (6.5%)	45 (11.4%)	45 (20.7%)	54 (35.1%)	
Diagnostic criteria for AKI							
Oliguria	19 (22.1%)	146 (30.6%)	201 (30.3%)	114 (28.9%)	65 (30%)	30 (19.5%)	<0.001
Anuria	11 (12.8%)	93 (19.5%)	155 (23.4%)	104 (26.4%)	49 (22.6%)	40 (26%)	
AKI achieves KDIGO-defined serum Creatinine elevation	22 (25.6%)	107 (22.4%)	152 (22.9%)	109 (27.7%)	59 (27.2%)	52 (33.8%)	
Others	34 (39.5%)	131 (27.5%)	155 (23.4%)	67 (17%)	44 (20.3%)	32 (20.8%)	
Timing of initiated CRRT *							
Early strategy	56 (65.1%)	279 (58.5%)	412 (62.1%)	225 (57.1%)	124 (57.1%)	85 (55.2%)	0.323
Delayed strategy	30 (34.9%)	198 (41.5%)	251 (37.9%)	169 (42.9%)	93 (42.9%)	69 (44.8%)	
Vital Sign							
Systolic BP (mmHg)	110 (104–123)	109 (102–122)	110 (101–120)	111 (102–120)	110 (101–125)	110 (99–121)	0.851
Diastolic BP (mmHg)	58 (52–66)	59 (52–66)	59 (52–68)	57 (51–65)	58 (52–67)	59 (51–67)	0.564
Pulse rate (bpm)	107 (88–122)	105 (88–117)	103 (88–117)	100 (85–115)	105 (90–117)	101 (87–114)	0.158
Body temperature (degree Celsius)	36.2 (35.5–37.2)	36.2 (35.4–36.9)	36.2 (35.4–36.8)	36.1 (35.4–36.8)	36.2 (35.6–36.9)	36.2 (35.7–36.9)	0.556
Respiratory rate (/min)	20 (17–24)	20 (17–23)	20 (17–23)	20 (17–22)	21 (18–23)	21 (18–25)	0.009
SPO_2_	98 (95–99)	97 (95–99)	97 (95–99)	97 (95–99)	97 (95–99)	96 (94–98)	0.118
Multiple organ support before CRRT—no. (%)							
Invasive mechanical ventilation	77 (89.5%)	423 (88.7%)	578 (87.2%)	333 (84.5%)	187 (86.2%)	127 (82.5%)	0.270
Extracorporeal Membrane Oxygenation (ECMO)	10 (11.6%)	73 (15.3%)	92 (13.9%)	30 (7.6%)	12 (5.5%)	4 (2.6%)	<0.001
Vasopressors support with norepinephrine or epinephrine	70 (81.4%)	400 (83.9%)	547 (82.5%)	332 (84.3%)	184 (84.8%)	129 (83.8%)	0.942
Medication use before CRRT—no. (%)							
Sedative	54 (62.8%)	358 (75.1%)	482 (72.7%)	284 (72.1%)	146 (67.3%)	111 (72.1%)	0.136
Corticosteroids	49 (57%)	264 (55.3%)	372 (56.1%)	225 (57.1%)	129 (59.4%)	94 (61%)	0.807
Loop diuretic	86(100%)	477(100%)	663(100%)	394(100%)	217(100%)	154(100%)	--
Parental nutrition	79 (91.9%)	401 (84.1%)	537 (81.0%)	330 (83.8%)	178 (82.0%)	144 (93.5%)	0.002
Furosemide	47 (54.7%)	237 (49.7%)	307 (46.3%)	206 (52.3%)	105 (48.4%)	83 (53.9%)	0.292
Antibiotics	84 (97.7%)	459 (96.2%)	619 (93.4%)	369 (93.7%)	210 (96.8%)	146 (94.8%)	0.113
Insulin use within 24 h after initial CRRT	36 (41.9%)	254 (53.2%)	355 (53.5%)	230 (58.4%)	130 (59.9%)	108 (70.1%)	<0.001
Urine Output before CRRT—ml/24 h	25 (6–52)	16 (4–35)	10 (2–29)	10 (3–29)	6 (0–22)	14 (2–42)	<0.001
Coexisting conditions—no. (%)							
Hypertension	20 (23.3%)	170 (35.6%)	260 (39.2%)	152 (38.6%)	81 (37.3%)	59 (38.3%)	0.101
Diabetes Mellitus	24 (27.9%)	149 (31.2%)	248 (37.4%)	162 (41.1%)	79 (36.4%)	53 (34.4%)	0.030
Hyperlipidemia	10 (11.6%)	67 (14%)	122 (18.4%)	71 (18%)	37 (17.1%)	31 (20.1%)	0.221
Coronary artery disease	12 (14%)	100 (21%)	171 (25.8%)	107 (27.2%)	52 (24%)	33 (21.4%)	0.050
Congestive heart failure	4 (4.7%)	51 (10.7%)	145 (21.9%)	95 (24.1%)	40 (18.4%)	26 (16.9%)	<0.001
Chronic pulmonary disease	8 (9.3%)	80 (16.8%)	132 (19.9%)	72 (18.3%)	52 (24%)	29 (18.8%)	0.056
Chronic renal disease	24 (27.9%)	145 (30.4%)	247 (37.3%)	162 (41.1%)	91 (41.9%)	64 (41.6%)	0.002
Malignancy	14 (16.3%)	87 (18.2%)	91 (13.7%)	73 (18.5%)	35 (16.1%)	32 (20.8%)	0.163
Cardiac arrhythmia occurrence at the baseline	8 (9.3%)	82 (17.19%)	136 (20.51%)	88 (22.34%)	52 (23.96%)	32 (20.78%)	0.033
Cardiac arrhythmia occurrence during the 24 h CRRT	6 (6.98%)	74 (15.51%)	124 (18.7%)	69 (17.51%)	38 (17.51%)	20 (12.99%)	0.073
Laboratory data before CRRT							
Albumin, mg/dL	1.9 (1.5–2.4)	2.1 (1.7–2.6)	2.3 (1.9–2.8)	2.3 (1.8–2.9)	2.2 (1.7–2.8)	2.2 (1.7–2.6)	<0.001
Hemoglobin, g/dL	9 (9–11)	10 (9–11)	10 (9–12)	9 (8–11)	10 (8–11)	10 (8–11)	0.022
WBC count, 1000/uL	9 (5–13)	11 (7–18)	12 (7–17)	12 (8–18)	13 (9–21)	13 (8–20)	<0.001
Platelet count, 1000/uL	74 (33–121)	95 (58–170)	109 (58–181)	111 (64–183)	136 (72–208)	141 (79–228)	<0.001
pH	7.3 (7.2–7.4)	7.3 (7.3–7.4)	7.3 (7.3–7.4)	7.3 (7.2–7.4)	7.3 (7.2–7.4)	7.3 (7.2–7.4)	0.003
Sodium, mmol/L	139 (133–145)	139 (134–143)	138 (134–142)	138 (134–142)	137 (134–141)	137 (133–142)	0.103
Lactate, mmol/L	5.1(1.6–9)	4(1.9–8.5)	4.1(1.7–8.7)	3.6(1.8–8.5)	3.5(2.1–8.1)	5.4(1.6–9.7)	0.737
Calcium, mg/dL	7 (7–8)	8 (7–8)	8 (7–8)	8 (7–9)	8 (7–8)	8 (7–9)	<0.001
Base Excess, mmol/L	−9 (−13–−6)	−8 (−11–−5)	−7 (−11–−4)	−8 (−11–−4)	−8 (−11–−5)	−8 (−12–−4)	0.235
O_2_ Saturation, %	99 (97–100)	98 (96–100)	98 (96–100)	98 (96–100)	98 (96–100)	98 (95–100)	0.220
Creatinine, mg/dL	2.0 (1.0–3.1)	1.9 (1.2–3.3)	2.5 (1.4–4.3)	2.4 (1.3–5.2)	2.8 (1.4–5.1)	2.7 (1.3–4.9)	<0.001
Outcome							
72 h mortality	27 (31.4%)	101 (21.2%)	147 (22.2%)	117 (29.7%)	82 (37.8%)	78 (50.6%)	<0.001
28-day mortality	52 (60.5%)	246 (51.6%)	353 (53.2%)	234 (59.4%)	137 (63.1%)	107 (69.5%)	<0.001
90-day mortality	59 (68.6%)	295 (61.8%)	431 (65%)	276 (70.1%)	158 (72.8%)	112 (72.7%)	0.014
In-hospital mortality	59 (68.6%)	302 (63.3%)	435 (65.6%)	276 (70.1%)	160 (73.7%)	114 (74.0%)	0.024

* Early-strategy group was defined by the renal-replacement therapy and was initiated within 24 h after the admins hospital documentation of failure-stage acute kidney injury.

**Table 2 jcm-11-03383-t002:** Adjusted hazard ratio for 72 h, 28-day, 90-day, and in-hospital mortality.

	72-h Mortality		28-Day Mortality		90-Day Mortality		In-Hospital Mortality	
	aHR (95% CI) ^a^	*p* Value	aHR (95% CI) ^b^	*p* Value	aHR (95% CI) ^c^	*p* Value	aHR (95% CI) ^d^	*p* Value
Serum (K+) level								
<3.0	1.10 (0.60–2.02)	0.756	1.10 (0.74–1.63)	0.630	0.99 (0.68–1.44)	0.968	1.24 (0.88–1.75)	0.228
3.0 to <3.5	Reference		Reference		Reference		Reference	
3.5 to <4.0	1.00 (0.73–1.38)	0.991	1.15 (0.93–1.41)	0.194	1.15 (0.95–1.39)	0.150	1.17 (0.97–1.39)	0.094
4.0 to <4.5	1.63 (1.17–2.26)	0.004	1.65 (1.32–2.07)	<0.001	1.66 (1.35–2.04)	<0.001	1.59 (1.31–1.94)	<0.001
4.5 to <5.0	2.12 (1.46–3.09)	<0.001	2.00 (1.52–2.64)	<0.001	2.06 (1.60–2.66)	<0.001	1.85 (1.44–2.36)	<0.001
≥5.0	2.52 (1.69–3.77)	<0.001	2.10 (1.54–2.86)	<0.001	1.96 (1.45–2.64)	<0.001	1.72 (1.30–2.27)	<0.001
Serum (K+) variability								
Q1: <0.35	1.03 (0.74–1.42)	0.870	1.10 (0.89–1.36)	0.381	1.07 (0.88–1.30)	0.490	1.12 (0.93–1.35)	0.242
Q2: 0.35 to <0.50	Reference		Reference		Reference		Reference	
Q3: 0.50 to <0.78	1.09 (0.81–1.46)	0.565	1.04 (0.86–1.27)	0.688	1.04 (0.87–1.24)	0.677	1.06 (0.89–1.26)	0.500
Q4: ≥0.78	1.61 (1.13–2.29)	0.009	1.39 (1.06–1.82)	0.018	1.43 (1.11–1.83)	0.005	1.31 (1.03–1.65)	0.027

Abbreviations: CI = confidence interval; cHR = crude hazard ratio; aHR = adjusted hazard ratio; aHR was calculated from multivariate Cox proportional regression model with a stepwise elimination procedure, and variables with a *p*-value < 0.05 in a univariate model were included in a multivariate model. Model 0: age, BMI, APACHE II at admission, strategy time, creatinine, O_2_ saturation, base excess, platelet count, albumin, lactate, calcium, sodium, hemoglobin, diabetes mellitus, hypertension, hyperlipidemia, chronic renal disease, malignancy, vasopressor, corticosteroids, parental nutrition, antibiotics, invasive mechanical ventilation, SPO_2_, respiratory rate, pulse rate, systolic BP, diastolic BP and body temperature ≥ 36.2 were commonly used confounders in mortality models. ^a^ adjusted for variables in model 0 plus insulin use. ^b^ adjusted for variables in model 0 plus furosemide. ^c^ adjusted for variables in model 0 plus chronic pulmonary disease, furosemide and insulin use. ^d^ adjusted for variables in model 0 plus chronic pulmonary disease and insulin use.

## Data Availability

Not applicable.

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
