# Peer review of "First 24-Hour Potassium Concentration and Variability and Association with Mortality in Patients Requiring Continuous Renal Replacement Therapy in Intensive Care Units: A Hospital-Based Retrospective Cohort Study"

_jcm, 2022, doi:10.3390/jcm11123383_

Round 1

Reviewer 1 Report

Mao IC et al studied the association between the blood K+ levels during the first 24 hours of CRRT in a wide cohort of ICU patients and in hospital mortality. They categorized the mean K+ levels (<3.0, 3.0 to <3.5, 3.5 20 to <4.0, 4.0 to <4.5, 4.5 to <5.0, and 5.0 mmol/L), and found J-shaped association between K+ levels and mortality, the lowest mortality being observed in the patients with mean K+ levels between 3.0 and 4.0 mmol/L. The risk of in-hospital death was significantly increased in those with the highest variability (HR and 95% CI = 1.31 [1.03–1.65] for inhospital mortality, 1.39 [1.06–1.82] for 28-day mortality, and 1.43 [1.11–1.83] for 90-day mortality).

They concluded that patients on CRRT may benefit from a lower serum K+ level and its tighter control, and the careful and timely correction of dyskalemia among these patients is crucial..

General comments

The paper is interesting. The Authors describe the relationship of K+ levels and mortality. However, this relationship was found during the first 24 hour of CRRT, and this should be underlined in the title, too. However, apart from the K+ levels, mortality was associated with many other variables, and the considered time of K+ correction is very short to influence mortality at 28-90 days.

Specific comment.

1) I suggest to consider the early mortality too, at 72 hours. Which was the relationship with K+ level and variability?

2 K+ levels influence the cell depolarization capacity. Was there any difference of the cardiac arrhythmia incidence at baseline and during the 24 hours of CRRT among the categorized K+ groups of patients?

3) Among the factors implicated in mortality rate there are the base Excess and lactate (see your Fig. 5). In detail, in group K+ <3.0 the median lactate levels were very high (5.1 mmol/L, see Table 1 on page 7, line 4 from the bottom). It is conceivable that in this group, with lowest pH and higher lactate, mortality was the result of these well known predictors of mortality, and that the low variability of K+ was the simple marker of not recoverable very deep hypokaliemia?

4) You wrote on page 12 line 305-306 “... Our results suggest that clinicians should correct dyskinesia more intensively to achieve normokalemic status”. But the directly correlation between K+ variability and mortality in all groups, but K+ <3.0, suggest a different conclusion,. In addition, in your data the group of patients who survived better was under the considered normal value of blood K+.

5) The variability of K+ also the result of dialytic technique. Which was the K+ concentration of dialysate during CRRT?

Reviewer 2 Report

I would probably try to resume a little bit better in order to make the article easier to read. Modalities emplloyed during CRRT should be described more accurately: modality, filter employed, renal dose and type of solutions employed for CRRT.

Round 2

Reviewer 1 Report

The work has improved a lot, in its conceptual part. I would have a further comment to improve the graphical presentation. Please enter the data in the various graphs in tables (Fig 1, Fig.2, Fig. 3, Tables 1 and 2) in chronological order, ie 72 hours, 28 days, 90 days.

Reviewer 2 Report

CRRT technique is not accurately described due to population heterogenicity which is easy to understand. However, authors should consider improving this part for future publications. Nice descriptive paper but no solid conclusions can be obtained if the technique is not registered. 
